



# Abrupt transitions in an atmospheric single-column model with weak temperature gradient approximation

Benjamin A. Stephens[1,2,3] and Charles S. Jackson[3]

[1]UWM Department of Mathematical Sciences, 3200 N Cramer St., Milwaukee WI 53211
[2]University of Texas Department of Physics, 2515 Speedway, C1600, Austin TX 78712
[3]University of Texas Institute for Geophysics, 10601 Exploration Way, Austin TX 78758

**Correspondence:** Benjamin A. Stephens (stepheba@uwm.edu)

**Abstract.** We document a feature of the tropical atmosphere that could be relevant to episodes of abrupt transitions in global climate that regularly occurred during the last ice age. Using a single-column model incorporating the weak temperature gradient (WTG) approximation, we find that abrupt transitions occur as the sea surface temperature is steadily increased. Because these transitions arise from the interplay of scales between local deep convection and the large-scale adjustments that are required to maintain weak temperature and pressure gradients, they are only present with the WTG approximation relevant for the tropics but may be of interest as a trigger for abrupt transitions in global climate. These transitions are marked by an abrupt change in the partitioning of rainfall between convective and large-scale (microphysics) subroutines, in addition to various other features of the column including cloudiness, vertical velocity, temperature, and humidity. We conclude that the transitions are initiated by a failure of evaporative cooling in the lower free troposphere. This leads to lower-column heating and a burst of convection that heats the upper free troposphere, increasing the large-scale rainfall rate and re-stabilizing the lower-column evaporative cooling.

## 1   Introduction

Explanations for abrupt climate change during the last ice age have largely focused on the role of the ocean, particularly the Atlantic meridional overturning circulation sensitivity to a freshening of the North Atlantic (Clark et al., 2002; Jackson et al., 2010). It is not yet clear whether this mechanism is sufficient to explain observed changes in tropical climate, particularly its hydrologic cycle and its monsoons (Peterson et al., 2000; Wang et al., 2001; F. W. Cruz et al., 2005; Weldeab et al., 2007; Clement and Peterson, 2008; Stager et al., 2011). Previous experiments with advanced climate models (Okumura et al., 2009) that have tested the freshwater forcing hypothesis do not contain the sensitivity required to explain the 40% reduction in atmospheric methane associated with cold stadials, an observation interpreted to reflect large reductions in tropical wetland extent (Brook et al., 1996, 2000; Fischer et al., 2008), although there may be complications in that interpretation (Kaplan, 2002; Kaplan et al., 2006). There is a need to explore more broadly how and why abrupt transitions in global climate have occurred, especially mechanisms within the tropical atmosphere (Wunsch, 2006; National Research Council, 2013).

Several studies document a relationship between sea surface temperature (SST) and either multiple equilibria or qualitative differences in rainfall behavior. Sobel et al. (2007) and Sessions et al. (2010) both find that rainy and dry states exist for





the same boundary conditions under the weak temperature gradient (WTG) approximation designed by Sobel and Bretherton (2000) to parameterize large-scale tropical dynamics for limited-domain models. Using a single-column model, Sobel et al. (2007) find the existence of dry and rainy states to be sensitive to both SST and the horizontal moisture advection rate, while Sessions et al. (2010) find similar behavior in a cloud-resolving model. These multiple equilibria have been understood in the context of explaining the spatial structure of tropical rainfall, but the fact that different equilibria exist under the WTG approximation is also interesting in the context of abrupt climate change. Held et al. (2007) also find that warmer SSTs are associated with qualitatively different types of rainfall behavior. Using a large, doubly-periodic, nonspherical and nonrotating domain run at GCM resolution with an initialization appropriate for the tropics, Held et al. (2007) document higher fractions of large-scale rainfall and the appearance of what they call "gridpoint storms"—areas sometimes spanning multiple grid cells generating predominantly large-scale (as opposed to parameterized convective) rainfall—for higher SSTs.

In this paper, we find that under a simple forcing, namely a continuous increase in SST, the single-column configuration of the Weather Research and Forecasting (WRF) model responds by abruptly transitioning to new configurations characterized by different fractions of large-scale rain, with important consequences for rainfall. Rather than rainy and dry states, however, our experiments demonstrate the existence of multiple equilibria in the relative amounts of convective and large-scale rainfall. These transitions only occur when the WTG approximation is implemented in the columns, making the transitions of interest in the tropics. Because the fraction of large-scale rainfall correlates strongly with circulation and spatial rainfall patterns, including in the response to $CO_2$ forcing (Stephens et al., 2019), understanding these transitions and what role they may play in global climate models is worthy of investigation.

## 2 Experimental Setup

The single-column model used in this study is the Weather Research and Forecasting (WRF) model version 3.5, compiled in single-column mode and modified by Wang and Sobel (2011) to implement the WTG approximation. SCMs are generally unable to account for large-scale dynamics, hence the WTG approximation is useful because it restores the coupling between convection and large-scale dynamics in the less computationally expensive SCM setting. Under the WTG approximation, the resolved vertical motion, rather than being set to zero or some specified velocity profile, is calculated to keep free-tropospheric temperatures close or equal to a temperature profile representing radiative-convective equilibrium (RCE). The justification for the WTG approximation lies in geostrophic balance, which for the tropical free troposphere implies small horizontal pressure and temperature gradients (Charney, 1963) due to gravity waves quickly eliminating pressure and temperature anomalies (Bretherton and Smolarkiewicz, 1989).

The equations governing the evolution of potential temperature $\theta$ and the water vapor mixing ratio $q$ in the WRF SCM with the WTG approximation are

$$\frac{\partial \theta}{\partial t} = W + Q_r^\theta + Q_c^\theta + Q_s^\theta + Q_m^\theta + Q_b^\theta, \tag{1}$$

$$\frac{\partial q}{\partial t} = -\omega \frac{\partial q}{\partial p} + Q_c^q + Q_s^q + Q_m^q + Q_b^q, \tag{2}$$





where all variables are functions of time $t$ and height $z$ (or pressure $p$). There are different ways to implement the WTG approximation in practice (for an "intercomparison" study exploring different methods and models for coupling convection to large-scale dynamics, see Daleu et al. (2015)). One approach is to assume that the tropical free-tropospheric temperatures do not
evolve in time at all (i.e. $\partial\theta/\partial t = 0$), but here the WTG approximation is implemented via "Newtonian relaxation", meaning instead of holding free-tropospheric temperatures fixed, they are continuously nudged back toward a target vertical temperature profile $\theta_{\text{RCE}}$ on some time scale $\tau$. $W$ in (1) represents the WTG Newtonian relaxation back to the RCE "background" profile $\theta_{\text{BG}}$,

$$W = -\frac{\theta - \theta_{\text{BG}}(z)}{\tau} = -\omega\frac{\partial\theta}{\partial p}. \tag{3}$$

As with the column resolution, a range of WTG relaxation time scales were tested, but the primary experiments were carried out with $\tau = 180$ min. The remaining forcing terms in (1) and (2) are from radiation (subscript $r$), the deep convective parameterization ($c$), the shallow convective parameterization ($s$), cloud microphysics ($m$), and the boundary layer parameterization ($b$). (As a practical matter, within the WRF code, the WTG forcing $W$ in (1) is combined with the boundary layer forcing $Q_b^\theta$, but they can be separated again later for analysis.) Eq. 3 allows the program to solve for the vertical pressure velocity $\omega$, which
is then used to evaluate the term $-\omega\dfrac{\partial q}{\partial p}$ in (2).

As mentioned above, it is possible to implement the WTG approximation such that the WTG term $W$ exactly cancels the other forcing terms and the change in temperature is zero. Here, because the WTG relaxation term is not constrained to exactly balance the diabatic forcing terms, the potential temperature at a given height or pressure level will depart from the background value in proportion to the total diabatic forcing at that level. Hence if $\sum Q$ were to abruptly increase at a given height, $\theta_S$ at that
height would abruptly increase as well, despite the WTG relaxation scheme. A further complication arises from the fact that the WTG approximation does not everywhere counter the diabatic heating, but only in the free troposphere—defined throughout this paper as levels above 850 mb. Below the free troposphere in the boundary layer, the vertical velocity is reduced linearly from its value at $p = 850$ mb to zero at the ground.

Rather than being wedded to specific parameterization schemes for radiation, microphysics, and so on as a typical GCM
would be, the more versatile WRF model can be run with a variety of physical parameterizations; this "menu" of physics options is one reason we chose to employ WRF for this study, since we can ultimately test any findings' sensitivity to the chosen parameterization schemes. However, because the immediate concern is to understand how low-level convergence can affect the tropical atmosphere at GCM resolution, here we carry out WRF-model experiments using the Community Atmosphere Model (CAM) physics parameterizations, including for radiation, microphysics, deep and shallow convection, and boundary layer
processes. A complete description of the WRF model version 3.5 can be found on the University Corporation for Atmospheric Research (UCAR) website. We note here an interesting observation, not explored in detail in this paper: the abrupt transitions documented below do not occur in the WRF model—even under the WTG approximation—unless the model is set to yield fractional cloud amounts. Certain configurations of WRF assign cloud fractions of zero or one only, presumably because WRF is often used for high-resolution modeling wherein grid cells are small enough to be either cloudy or not.





For radiation purposes, we set the column at a latitude of zero degrees. Because cloud-radiation feedbacks can complicate the interaction between convection and large-scale advection (which the WTG approximation is usually employed to study), WTG experiments often make use of prescribed radiative cooling, such that the column cools via emission of longwave radiation at a rate matching tropical observations. The WRF model as modified by Wang and Sobel (2011) includes this option of idealized cooling in the troposphere (they use a rate of $-1.5$ K/day). Because we are interested in how the standard CAM physics parameterizations behave under SST forcing, we use the realistic CAM radiation scheme in our primary experiments, though we have tested the forcing under prescribed radiation and comment on those results below. As Wang and Sobel (2011) note, ice clouds in the upper troposphere can block outgoing radiation in a realistic tropical setting, an effect they do not account for but which will be important in this study.

In testing these results' sensitivity to various model settings, a range of horizontal and vertical resolutions were ultimately used, but the standard SCM setup was for a 100-km horizontal resolution (intentionally similar to that of a typical GCM) and 50 vertical levels up to a height of 20 km. The time step was set to 5 minutes.

We use the fraction of large-scale rain generated by the model as a basic diagnostic. Like a typical global atmospheric model, the WRF model generates both convective and large-scale rainfall (the latter is usually called "non-convective" rain in the WRF context). In Stephens et al. (2019), $f_{LS}$ was defined as the tropical (30°S-30°N) mean large-scale rainfall rate divided by the tropical mean total rainfall rate $f_{LS} \equiv \sum P_{LS} / \sum (P_{LS} + P_C)$, where the total convective rain rate $P_C$ included shallow convective rainfall. Since here we are using a simpler one-dimensional model that does not automatically sum deep and shallow convective rain, $f_{LS}$ will be defined as

$$f_{LS} \equiv \frac{P_{LS}}{P_{LS} + P_D + P_{SH}}, \tag{4}$$

where $P_{LS}$, $P_D$, and $P_{SH}$ are the SCM large-scale, deep convective, and shallow convective rainfall rates respectively.

The SST-forcing experiments analyzed in this paper were all done similarly. Using 300 K as a typical tropical SST, the WRF SCM is first run to radiative-convective equilibrium with this surface temperature over a period of 180 days. The final thirty days of this experiment are then averaged to extract equilibrated pressure, temperature, height, and humidity profiles, which are then used to determine the background profile $\theta_{BG}$ for the WTG routine (Eq. 3), and then the experiments are started again at 300 K with the WTG approximation active. After an initial 90-day startup, the SST is continuously increased at a fixed rate (usually 0.5 K/month; varying the rate of SST increase or decrease does not have a strong effect on the SCM's behavior) for sufficient time to reach a specified maximum temperature (usually 304 K). The temperature is then held at the maximum value for a period of 90 days, and then continuously reduced back to 300 K where the SST is held for a final 90-day period. The experiments are all initialized with the same temperature and moisture profiles consistent with a SST of 300 K.

## 3 Findings

This section is divided into four subsections. In the first, we describe general observations we have made using the WRF SCM with and without the WTG approximation, including the rainfall behavior for both cases and how the column balances the





temperature and moisture forcings in each case. In the second subsection, we document the hysteresis and multiple equilibria
we have found within the column under the WTG approximation with SST forcing, and in the third subsection we describe
in greater detail the abrupt transitions that have been our primary focus (to be further analyzed in the following Discussion
section). In the final subsection, we briefly describe a unique state, characterized by $f_{\mathrm{LS}} = 1$, into which the column sometimes
abruptly transitions and from which it never seems to recover.

## 3.1   RCE vs. WTG experiments: general observations

We begin by describing the typical features of our WRF SCM radiative-convective equilibrium experiments and how these
experiments are typically affected by activating the WTG approximation *without* anomalous SST forcing. Figure 1 displays
tropical temperature, humidity, and cloud profiles averaged from the last 30 days of two 180-day WRF SCM experiments,
one a radiative-convective equilibrium experiment and one using the WTG approximation. The temperature plot includes
ERA-Interim reanalysis data for reference. Figure 1 shows several important effects of the WTG approximation: along with
significantly increasing the free tropospheric moisture, it induces strong large-scale upward motion and hence condensation
causing large cloud fractions in the upper column. The warming effect of this condensation above $\sim$700 mb is evident in the
temperature sounding for the WTG experiment, although the highest parts of the column show cooling. While the large cloud
fractions are unrealistic, they are a persistent feature of WTG experiments using the (realistic) CAM radiation parameterization.

The way the column balances the various forcing terms of (1) and (2) changes dramatically when the WTG approximation
is active. In the RCE case (Figure 2, left column), the dominant $\theta$-forcing balance over a large part of the column is between
convective heating and radiative cooling, and (1) becomes $Q_r^\theta \approx Q_c^\theta$. Meanwhile, the $q$-forcing terms are small above roughly
800 mb, with the convective and boundary layer schemes balancing each other below that level. With the WTG approximation
active (Figure 2, right column), the balances are qualitatively and quantitatively different. Qualitatively, the WTG column now
shows two different kinds of balance for the upper and lower troposphere. Above roughly 600 mb (the height of the cloud base),
the dominant balance is between heating from condensation (microphysics) and cooling from the WTG relaxation, $W \approx Q_m^\theta$. In
the lower column, the dominant balance is between convective heating and evaporative cooling (also microphysics), $Q_m^\theta \approx Q_c^\theta$,
with shallow convection and the WTG relaxation playing more minor roles. Radiation plays a comparatively minor role in the
WTG case, a consequence of the noted extreme cloudiness of the upper column; infrared radiation from below is absorbed at
the cloud base and longwave radiative cooling dominates at the top of the column, while shortwave radiation from space is
almost all absorbed by high-level clouds (Table 1). Because of this, evaporation from the microphysics parameterization is the
only available source of cooling in the lower column and will play a critical role in the abrupt transitions described below. As
the lower panels of Figure 2 shows, the moisture forcings largely follow the $\theta$-forcings in the WTG case, and indicate the much
more important role played by moisture when the WTG approximation is active.

Quantitatively, the magnitudes of the forcings are much larger with the WTG approximation active, likely a consequence of
a positive heating feedback active in the upper column (discussed further in the Discussion section). Of particular interest in
this connection is the fact that while mixing via deep convection is now largely restricted to $p \gtrsim 650$ mb—probably because
of the reduced lapse rate and hence stabler profile near that level (see Figure 1)—the deep convective $\theta$-forcing is much larger





than in the RCE experiments. The greater convective heating cannot be attributed to greater CAPE, which is much larger in the RCE case, wherein the upper column is cooler. Rather, the greater deep convective $\theta$-forcing is likely due to the greater abundance of moisture and therefore larger condensational heating as convective plumes rise into cooler air.

Considering the very different temperature and moisture forcing balances within the column, it is unsurprising that the RCE and WTG cases show different rainfall behavior, both in rainfall rates and type (Table 1). Using the CAM physics options, a typical $f_{LS}$ for the standard WRF SCM is about 0.3, while with the WTG approximation active $f_{LS}$ is generally larger due to the greater upper-column microphysics activity.

The column response to increasing SST is very different with and without the WTG approximation active. In the RCE case, an increasing SST heats the column and leads to greater intensity of rainfall while driving the fraction of large-scale rain down slightly (Figure 4). On the other hand, with the WTG approximation active, as SST increases the column begins to show stepwise abrupt transitions to larger $f_{LS}$, a robust outcome of such experiments (apart from the occasional $f_{LS} \to 1$ behavior). Consistent with the results displayed in Figure 3, when the WTG approximation is inactive, $f_{LS}$ is seen to decrease somewhat as the SST is increased and increase again as the SST is lowered. From Figure 3, it is apparent that when the column is run without the WTG approximation, higher SSTs result in greater convection, but with the WTG approximation active, the presence of a large-scale vertical pressure velocity $\omega$ allows the column an additional means of handling of the upward redistribution of energy.

## 3.2 Hysteresis and multiple equilibria

Figure 4 confirms that with the WTG approximation active, the column exhibits hysteresis: the evolution of $f_{LS}$ as the column warms is different from its evolution as the column cools. To document the implied multiple equilibria, we carried out a modified version of the experiment described above, wherein we paused the SST increase/decrease at a specified "resting" SST and allowed the model to run for 30 days.

Table 2 and Figure 5 document two different SCM solutions averaged over those 30-day periods, which we call E1 and E2, for the same resting SST of 304.5 K (in this particular set of experiments, the SST was increased from 301 K to 305 K, but the qualitative behavior is similar to our standard 300 K-to-304 K experiments). E1 is the solution obtained during the warming phase of the experiment, and E2 is obtained during the cooling phase. It is evident from Figure 5 that in the case of E2, the model has settled into a warmer, wetter solution with stronger vertical motion, although there is nuance in how the model achieves balance between heating and cooling in this case relative to E1. The clearest difference is in the shallow convective subroutine, where for E2 shallow convection plays a much greater role in heating the column between roughly 900-600 mb while the rate of shallow rainfall $P_{SH}$ (while small compared to deep convective or large-scale rain) more than doubles (Table 2). Both convective parameterizations also reach slightly higher into the column. E2's enhanced shallow convective heating is offset between 900-650 mb by both the WTG relaxation and evaporative cooling from the microphysics scheme—it is noteworthy that the microphysics parameterization is unable to balance the lower-column heating alone. Near 600 mb, however, E2 shows a net increase in heating from shallow convection. The deep convection $q$-forcing shows that the E2 solution is furthermore removing more vapor from below 600 mb and depositing slightly more above.



### 3.3 Abrupt transitions and quasi-stationary states

The results described in this section and analyzed in the Discussion section below are taken from an RCE experiment at a SST of 300 K and a SST-forcing experiment increasing the temperature from 300 K to 304 K with a rate of increase of 0.5 K/month. The average $f_{\mathrm{LS}}$ for the last 30 days of the 180-day RCE experiment is 0.28. The WTG experiment first equilibrates with an $f_{\mathrm{LS}}$ of about 56.0%, but as the SST is increased, the column shows three abrupt transitions to higher $f_{\mathrm{LS}}$, roughly 62.7%, 64.5%, and finally 66.1% (Figure 6, upper left panel). As noted, the column shows additional abrupt behavior and hysteresis as SST is decreased, but a close analysis of the cooling phase is left for future work.

Figure 6 and Table 3 show both the evolution of $f_{\mathrm{LS}}$ as the WRF SCM heats up and the major characteristics of the four quasi-stationary states (S1, S2, etc.) observed before/after the abrupt transitions. It is clear from the profiles of $\omega$, the various $\theta$- and $q$-forcing terms, and the rain and snow mixing ratios and number concentrations that, in general, the magnitudes of upward motion, heating, moistening, and rainfall grow as the surface temperature increases. There are, however, some features that show interesting qualitative changes as one state gives way to another. In particular—and as noted above in discussing the multiple equilibria at 304.5 K—the shallow convection profile shows the most marked change, with both the magnitude and shape of its profile changing from state to state, with progressively greater activity higher above the surface. Moreover, both the deep and shallow convective profiles reach higher into the column for higher SST. (Consistent with this observation, the cloud base moves higher with each transition.)

Before looking closely at the abrupt transitions, it is worth noting some general features of the column evolution leading up to the transitions, evident in the left panel of Figure 7. As expected, temperatures near the surface begin to increase along with the SST forcing, but, for example, the temperature one level above the surface increases at only 0.15 K/month, much slower than the 0.5 K/month SST increase. (For comparison, in an identical SST-forcing experiment without the WTG approximation, the temperature one level above the surface increases at 0.46 K/month.) Also as expected, given the way the WTG approximation is designed to operate, as the height approaches $p = 850$ mb where the WTG relaxation becomes active, temperatures are more stable. More surprising, however, is the column behavior above roughly 650 mb. Here, temperatures increase even more quickly than near the surface (e.g., at a rate of roughly 0.35 K/month at $p \approx 460$ mb, with some higher levels showing even larger warming rates). This can only be an effect of the convective parameterizations removing heat from near the surface and moisture from throughout the lower column, and depositing both near 650 mb (see Figures 2 and 6), where the large-scale advection generated by the WTG relaxation can "take over", carrying this moisture aloft into cooler air, where the microphysics generates the observed condensational heating. Moreover, convection is delivering this heat and moisture to the upper column at a growing rate—if the rate were constant, the WTG relaxation could stabilize the temperatures. This increasing rate of heat export from near the surface is probably consistent with the fact that the surface is heating so much more slowly than it would in the absence of the WTG relaxation. This general behavior causes the more stable "middle" part of the column, between roughly 900-650 mb, to grow increasingly out of sync with the regions heating above and below.

A close look at the abrupt transitions shows some features common to all. First, the forcing that most closely follows the lower-column heating is the microphysics, and within the microphysics routine it seems clear that a loss of evaporative





cooling is the main driver of the rapid temperature increase at the transition. In the standard model output, this relationship is
especially clear during the first transition (Figure 8). And while the microphysics forcing grows noisier as SST rises, making the
relationship slightly less clear in the standard output for the second and third transitions, unprocessed, high-resolution output
obtained from the microphysics routine confirm the same pattern for the second and third transitions as well (not shown).
As the evaporation starts to fail and temperature starts to increase, a new positive feedback develops: the WTG relaxation
responds with cooling and stronger upward motion, advecting moisture upward and causing the mixing ratio and relative
humidity to rapidly increase alongside temperature, despite the falling evaporation and likely exacerbating the evaporation
shortfall. Moreover, the WTG relaxation plays a progressively greater cooling role in the lower column with each transition,
while the microphysical cooling recovers but does not gain much ground between roughly 700-900 mb over the course of the
SST forcing (see Figure 6). It is possible that generally high relative humidity in the lower column limits evaporative cooling,
such that as the column warms the microphysics ultimately cannot provide enough cooling to balance the heating from the
convective routines. Growing local relative humidity, and therefore less ability to take up additional water vapor, might even
be suggested as a trigger for the transition, but the relative humidity does not increase markedly before the first transition, and
is smoothly *decreasing* before the onset of the second and third transitions, most likely because the convective routines are
quickly removing moisture from the lower column.

Second, the upper and lower parts of the column experience the transitions differently. Visible in the left plot of Figure 7
but shown more clearly in Figure 9, over the few days during which a transition takes place, the upper-column temperatures
temporarily stabilize (or even slightly decrease) while the lower-column temperatures show a rapid but fairly smooth increase.
Consistent with this, the upper-column diabatic heating and WTG relaxation also temporarily stabilize or reverse their trends.
Toward the end of the rapid temperature and moisture increase in the lower troposphere, the corresponding upper-column
variables abruptly transition to new values. For clarity and simplicity, we will refer to these two types of transition behavior as
"rolling" for the lower column and "snapping" for the upper column.

Third, it seems clear that the transition's transmission to the upper column and termination are closely connected to the
convective parameterizations: the convective scheme's $\theta$- and $q$-forcings briefly spike near the end of the lower-column "roll",
just as the upper-column variables "snap" into their new quasi-stationary values (see the right panel of Figure 7, which shows
the upper-column temperature evolution along with the abrupt increase in deep convective $\theta$-forcing near 600 mb at the "snap").
These are among the rare occasions when the convective routine is able to penetrate above $\sim$600-650 mb, and after this burst
of convection near the end of each transition, convective mixing reaches (usually one level) higher into the column than before.
Aside from these spikes at the transitions, the deep convective forcings grow quite linearly with SST.

Finally, while most variables follow the behavior of their corresponding part of the column (i.e. most lower-column variables
show rolling behavior, while most upper-column variables show snapping behavior), some variables do not. The convective
routines are one example, but the rain and snow mixing ratios (determined by the microphysics parameterization) also break
the pattern, showing snapping behavior even in the lower column (Figure 9, right panel). This is consistent with precipitation
changes initiated in the upper column subsequently becoming apparent in the lower column as the rain and snow precipitate
out.





In some cases, there are sudden changes in certain variables prior to the transitions, although a causal relationship is not
clear. The most intriguing of these "precursor" events involve abrupt changes in the mixing ratios and number concentrations
of raindrops (lower column) and/or snowflakes (upper column). These shifts may signal threshold-crossing behaviors in the
microphysics subroutine as it responds to the heating environment. Figure 10 gives an example of this type of precursor
behavior for the second and third abrupt transitions. In the left panel, the raindrop number concentration $N_r$ is shown against
the temperature evolution; $N_r$ clearly shifts abruptly at $t \approx 180$ days, again at the first temperature transition, and then there is
another possible shift at $t \approx 275$ days before the second temperature transition. In the right panel of Figure 10, the rain water
mixing ratio $q_r$ is shown against the water vapor mixing ratio. Again, it appears $q_r$ shifts near $t \approx 180$ days, although a shift is
less evident near $t \approx 275$ days. However, these plots are for one level only (around $p \approx 835$ mb) and are not representative of
the entire column. The microphysics variables show rich behavior that is often difficult to interpret—no doubt a consequence of
the richness of the microphysics routine, described briefly in the next section—but it is worth noting that the changes depicted
in Figure 10 do seem to coincide with the initiation of a slow temperature increase leading up to the transitions, clearer in the
second and especially third transition than in the first. This may imply correlation or causation but caution is warranted.

### 3.4  The $f_{\mathrm{LS}} \to 1$ case

Under certain conditions, the WRF SCM under the WTG approximation can transition into a state with $f_{\mathrm{LS}} = 1$. In this state the
convective parameterizations shut down completely, the column becomes cloudy almost top to bottom, the $\theta$- and $q$-forcings
grow to even larger magnitudes, and the lower-column pressure velocity $\omega$ becomes much larger than usual. Furthermore, once
the column enters this state, it seems permanent; the column seems never to recover from the $f_{\mathrm{LS}} \to 1$ transition even when the
SST is decreased again.

Figure 11 and Table 4 give average values for a number of variables for both a typical quasi-stationary state (S1) and the
$f_{\mathrm{LS}} = 1$ state (S2) in an experiment showing the $f_{\mathrm{LS}} \to 1$ transition just before the end of the SST increase. This experiment is
identical to the experiment analyzed in Section 3.3, except that the SST forcing is from 301.15 K (28°C) to 305.15 K and the
WTG background profile is calibrated to SST 301.15 K.

Convective profiles are omitted from Figure 11 as they are zero for the $f_{\mathrm{LS}} = 1$ state. Figure 11 also shows both the WTG
relaxation forcings and the pure boundary layer forcings. Summed together, the cooling from the WTG and boundary layer $\theta$-
and $q$-forcings balance the heating from microphysics. The boundary layer/moist turbulence scheme acts higher in the column
than usual, where it appears to play a role in mixing heat and moisture across the cloud base layer, similar in some ways to the
function previously performed by the convective parameterizations.

As is evident from Table 4, convective rainfall not only ceases for $f_{\mathrm{LS}} = 1$, but the microphysics forcings and associated
large-scale rainfall show dramatic increases, with forcings and rainfall rates an order of magnitude larger than those obtained
for lower $f_{\mathrm{LS}}$, and as much as two or three orders of magnitude larger than the forcings and rates for a typical RCE experiment
(see Figure 2). Moreover, the balance of forcings in the column changes qualitatively again, with the WTG relaxation and
boundary layer scheme now working together to balance microphysical heating (via melting or condensation) over the full
depth of the column. Evaporation has now failed completely to cool the lower column and has switched over to warming.





## 4  Discussion

It seems clear that lower-column evaporative cooling plays an essential role both in initiating the abrupt transitions and keeping
the column from falling into the $f_{\mathrm{LS}} \to 1$ state. To better understand this, however, it is worth considering the overall balance
of the column when the WTG approximation is active. We noted above that the magnitudes of the WRF SCM $\theta$-forcings are
much larger in the WTG case than in the RCE case. This leads to a consideration of how the column stabilizes itself at the
beginning of the WTG experiments, a close look at which suggests a positive heating feedback active (at first) in the upper
column which is ultimately balanced by a series of diabatic forcings (Figure 12). Because heating in the upper column (whether
coming from the convective or microphysics parameterizations) can only be balanced here by radiative cooling or the WTG
relaxation, but more efficiently by the latter, heating at high levels generates an upward large-scale motion by (3), which then
generates more heating due to condensation or freezing as water vapor is lifted into cooler air. The resulting cloudiness also
blocks part of the outgoing infrared radiation from lower in the column, causing heating near the cloud base and causing the
feedback to reach lower into the column. This feedback, which begins near the top of the column, causes upward motion lower
and lower in the column, until the falling precipitation reaches levels at which its melting or evaporation generates enough
cooling to stop the feedback mechanism from reaching even lower into the column. Meanwhile, the large-scale rain resulting
from this upper-column activity generates proportionally greater evaporative cooling below as the large-scale rain falls through
the lower column, with this cooling balanced in turn by enhanced heating from the convective parameterizations. Finally,
enhanced convection (especially coming from the shallow scheme) is able to deliver moisture and heat from the lower column
to the layers around the cloud base, feeding the upper-column microphysics routine and thus allowing the balances depicted in
Figure 12 to hold for an overall warmer column.

Some temperature forcings can respond quickly and without strong limits to temperature changes, while other forcings are
limited such that the column equilibrium could become strained. The WTG relaxation simply adds or removes heat from the
system (while inducing a vertical velocity), and has no inherent limitations. The shallow convective parameterization too is
unlimited in its vertical reach and precipitation rate; indeed, it can represent deep convection on its own if no deep convective
parameterization is employed (Park and Bretherton, 2009). It seems likely that this ability of the shallow scheme to work
well beyond its "shallow" capacity is responsible for the proportionally greater increase in its heating role as SST increases
(Figure 6). On the other hand, the deep convection scheme is limited by the CAPE-consumption time scale; it consumes
CAPE at a rate proportional to the amount of CAPE (Zhang and McFarlane, 1995). Upper-column microphysical heating is
limited by the availability of moisture there—if the convective routines and WTG lifting cannot deliver enough moisture, the
condensational heating will not be maintained.

Similarly, lower-column evaporation is limited by the abundance of rainfall and its microphysical characteristics coming
from above and by the local relative humidity. Because evaporative cooling provided to the lower column by the microphysics
parameterization appears to be the first element of the Figure 12 balance to fail as the system transitions to each new state—and
because the lower-column heating closely follows the reduced evaporation—we attribute the transitions to nonlinearities within
the microphysics routine.



The CAM microphysics routine is due to Hugh Morrison and various collaborators (Morrison et al., 2005; Morrison and Gettelman, 2008), and is based on diagnostic equations for the number concentration $N$ and mixing ratio $q$ of rain or snow droplets (but here the subscript $r$ is for rain):

$$\frac{1}{\rho}\frac{\partial}{\partial z}\left(\rho V_N N_r\right) = -\sum_i \left(\frac{\partial N_r}{\partial t}\right)_i \tag{5}$$

$$\frac{1}{\rho}\frac{\partial}{\partial z}\left(\rho V_q q_r\right) = -\sum_i \left(\frac{\partial q_r}{\partial t}\right)_i, \tag{6}$$

where $V_N$ and $V_q$ are the terminal fall speeds for rain (or snow) weighted by number and mass respectively, and $i$ indexes a number of terms due to various microphysical processes (condensation, melting, droplet growth by accretion or self-interaction, etc.), among which is evaporation of falling rain. Evaporation of rain in the CAM microphysics, of primary interest here, is given by

$$P_E = \frac{2\pi N_0(S-1)\left[0.78\Lambda^{-2} + 0.32 S_C^{1/3} a^{1/2}\nu^{-1/2}\Gamma(\frac{b+5}{2})\Lambda^{-(b+5)/2}\right]}{\dfrac{L^2}{KR_vT^2} + \dfrac{R_vT}{e_sD_v}}. \tag{7}$$

The numerator of (7) is related to the amount of available moisture while the terms in the denominator account for evaporation's thermal and diffusive effects: $N_0$ and $\Lambda$ are spectral parameters determining the raindrop size distribution [these are functions of $N_r$ and $q_r$ from (5) and (6)]; $a$ and $b$ are empirically determined parameters giving the raindrop terminal fall velocity; $S \equiv e/e_s$ is the ratio of ambient vapor pressure to saturation vapor pressure; $\nu$ is the kinematic viscosity; $K$ and $D_v$ are the thermal conductivity and mass diffusivity of water vapor in air respectively; and $S_C \equiv \nu/D_v$ is the Schmidt number. (The thermal conductivity can be written as $K = c_p\rho\kappa$, where $\kappa$ is the thermal diffusivity. In the WRF code, $\kappa$ is replaced by the mass diffusivity $D_v$, acceptable for air-water vapor systems which have a Lewis number (Le $\equiv \kappa/D_v$) approximately one.) The temperature- and moisture-dependence of (7) is quite complicated, with most of the variables determined by temperature and/or moisture.

As mentioned above, raw, high-resolution output taken from this equation during the WTG experiment confirms that the magnitude of $P_E$ diminishes with each lower-column rolling temperature increase. Moreover, analysis of the individual terms of (7) indicates that the numerator is stable during the onset of the transitions, an expected result considering that the raindrop mixing ratio $q_r$ and number density $N_r$—related to terms in the numerator—transition with the upper-column "snap" rather than with the lower-column "roll." The saturation vapor pressure and thermal and diffusive terms, however, show rolling behavior alongside the temperature change, indicating that the local environment is more important in initiating the transition— that is, evaporation is not keeping pace with local heating.

The $f_{LS} \to 1$ behavior can be understood as a "runaway" case of the positive WTG heating feedback. If the only way in which the column can balance heating is via the WTG relaxation, this will generate additional heating above via lifting (and condensation) of moisture, and heating below via absorption of infrared radiation at the newly generated cloud base. Indeed, a preliminary look at the $\theta$-forcings in several experiments showing the $f_{LS} \to 1$ transition reveals that near the transition, there is a point at which the evaporative cooling from the microphysics routine switches over into condensational heating. This means





the deep and shallow convective routines and the microphysics parameterization are all working to heat the column at these levels; at this point, the only way the column can balance this heating is via radiative cooling or the WTG relaxation, the latter

of which is again the more efficient. Hence the lower part of the column now experiences the WTG feedback, which spreads over the lower column leading quickly to $f_{\mathrm{LS}} \to 1$.

The extreme cloudiness shown by the WTG column, a consequence of using the realistic CAM radiation parameterization, would seem problematic. While cloud fractions can be fairly large in some areas of the tropics at high levels—for example, CALIPSO 2006-2014 data shows cloud fractions as high as $\sim 0.8$ for $p < 440$ mb in the area of the Maritime Continent—

the 100% cloud cover consistently observed over the upper half of the WRF SCM column with the WTG approximation is unrealistic (although even higher fractions may be possible at smaller length and time scales). Prescribed radiative cooling seems to alleviate this problem (at least in the absence of anomalous SST forcing), keeping cloud fractions similar to their RCE values (which are also closer to observations), but this option is designed to isolate and study the interaction between convection and large-scale vertical advection. Since we are pursuing a different question involving the realistic case including

cloud-radiation interactions, prescribed radiation is not an optimal solution. Furthermore, even with prescribed convection, SST forcing again generates the unrealistic 100% cloud fractions in the upper column.

However, an intriguing possibility is that the high cloud and stratiform rainfall fractions can be interpreted in the context of mesoscale convective systems (MCSs). In a historical review of MCSs, Houze (2018) notes that mature MCSs typically show distinct convective and stratiform rainfall regions, with convective below and stratiform above, and—particularly in

MCSs over the ocean—can have stratiform rainfall fractions as high as 70%. The heating profile of such systems is top-heavy according to the amount of stratiform rain. These MCS features are consistent with the behavior of the WRF SCM under the WTG approximation, leading one to speculate whether the WTG approximation may emulate some behavior of mature MCSs under certain conditions. There are some discrepancies: the stratiform regions described by Houze (2018) are weakly buoyant, whereas those appearing in our simulations show intense upward motion.

As noted in the introduction, these transitions are a robust feature of the WRF SCM with CAM physics, as long as the WTG approximation is implemented. Varying horizontal resolution has little effect on the transition behavior. Varying the WTG relaxation timescale $\tau$ has a more significant effect, with the $f_{\mathrm{LS}} \to 1$ behavior more common and happening earlier for larger $\tau$. Furthermore, we find broadly similar behavior with prescribed radiative cooling of $-1.5$ K/day over the troposphere. The steplike transitions are not as common, and $f_{\mathrm{LS}}$ can show more oscillatory behavior, but the column can still transition to higher

$f_{\mathrm{LS}}$ and the $f_{\mathrm{LS}} \to 1$ transition can still occur.

## 5  Conclusions

Using the WRF single-column model with CAM physics parameterizations and incorporating the weak temperature gradient approximation, we have found that abrupt transitions occur when SST is continuously increased, mimicking low-level convergence of sensible and latent heat. Beyond certain threshold temperatures, the column abruptly transitions to a new con-

figurations with larger fractions of large-scale (or non-convective) rainfall. The stability of the column under WTG conditions





appears to depend on delicate balances established between the various temperature and moisture forcings, and the SST-induced transitions appear to be initiated by a drop in the lower-column evaporative cooling coming from the microphysics parameterization. This breakdown can be traced to Eq. 7 which determines evaporative cooling; raw output from this equation suggests that the local temperature and moisture are more important than droplet number or mixing ratio in initiating the transitions.

*Data availability.* WRF model output used in generating the figures for this paper, along with input files for the WRF experiments conducted, have been made available at zenodo.org (Stephens and Jackson, 2020). The version of WRF used and cited in this paper are publicly available except for minor modifications, e.g. to continuously increase the sea surface temperature.

*Author contributions.* Stephens drafted the paper which was co-edited by Stephens and Jackson. The effort comprises a part of Stephens's Ph.D. thesis research with Jackson as adviser.

*Competing interests.* The authors have no competing interests to declare.

*Acknowledgements.* We extend thanks in particular to Adam Sobel and Shuguang Wang, whose version of the WRF model was used in this study, and to Vincent Larson and Benj Wagman for helpful comments. Funding for this project was provided by the US Department of Energy, Office of Science, Biological and Environmental Research Awards DE-SC0016401 and DE-SC0006985.



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





|  | $f_{\text{LS}}$ | $P_{\text{D}}$ | $P_{\text{LS}}$ | $P_{\text{SH}}$ | SWD | GLW | OLR | HFX | LH |
|---|---|---|---|---|---|---|---|---|---|
|  |  | mm/day | mm/day | mm/day | W/m$^2$ | W/m$^2$ | W/m$^2$ | W/m$^2$ | W/m$^2$ |
| RCE | 0.28 | 1.26 | 0.46 | 2.61e-4 | 270 | 358 | 272 | 8.94 | 99.3 |
| WTG | 0.56 | 9.55 | 12.1 | 2.0e-3 | 14.6 | 416 | 34.9 | 48.2 | 165 |

**Table 1.** A series of quantities averaged over the last 30 days from 180-day RCE and WTG approximation experiments without anomalous SST forcing. After $f_{\text{LS}}$, the three types of precipitation (deep convective $P_{\text{D}}$, large-scale $P_{\text{LS}}$, and shallow convective $P_{\text{SH}}$) are shown, followed by the shortwave radiation to the surface (SWD), the outgoing longwave radiation from the ground (GLW), the outgoing longwave radiation at the top of the column (OLR), and the surface sensible (HFX) and latent (LH) heat fluxes.





| | $f_{LS}$ | $P_D$ | $P_{LS}$ | $P_{SH}$ | SWD | OLR | HFX | LH |
|---|---|---|---|---|---|---|---|---|
| | | mm/day | mm/day | mm/day | W/m$^2$ | W/m$^2$ | W/m$^2$ | W/m$^2$ |
| E1 | 0.61 | 42.4 | 66.0 | 0.026 | 12.9 | 39.9 | 67.3 | 267 |
| E2 | 0.63 | 45.4 | 77.8 | 0.056 | 11.4 | 40.7 | 57.3 | 231 |

**Table 2.** Average values for a series of quantities for the two distinct SCM equilibria (E1 and E2) shown in Figure 5. After $f_{LS}$, the three types of precipitation (deep convective $P_D$, large-scale $P_{LS}$, and shallow convective $P_{SH}$) are shown, followed by the shortwave radiation to the surface (SWD), the outgoing longwave radiation at the top of the column (OLR), and the surface sensible (HFX) and latent (LH) heat fluxes.





| | $f_{LS}$ | $P_D$ | $P_{LS}$ | $P_{SH}$ | SWD | GLW | OLR | HFX | LH |
|---|---|---|---|---|---|---|---|---|---|
| | | mm/day | mm/day | mm/day | W/m$^2$ | W/m$^2$ | W/m$^2$ | W/m$^2$ | W/m$^2$ |
| S1 | 0.56 | 9.58 | 12.2 | 2.1e-3 | 17.6 | 416 | 35.6 | 48.3 | 165 |
| S2 | 0.63 | 14.6 | 24.5 | 5.3e-3 | 13.3 | 412 | 36.5 | 58.5 | 207 |
| S3 | 0.64 | 19.5 | 35.4 | 14.8e-3 | 14.0 | 416 | 38.0 | 63.7 | 235 |
| S4 | 0.66 | 23.4 | 45.7 | 31.9e-3 | 13.4 | 422 | 39.5 | 63.2 | 243 |

**Table 3.** Average values for a series of quantities for the four states from the experiment shown in Figure 6. After $f_{LS}$, the three types of precipitation (deep convective $P_D$, large-scale $P_{LS}$, and shallow convective $P_{SH}$) are shown, followed by the shortwave radiation to the surface (SWD), the outgoing longwave radiation from the ground (GLW), the outgoing longwave radiation at the top of the column (OLR), and the surface sensible (HFX) and latent (LH) heat fluxes.





|  | $f_{\mathrm{LS}}$ | $P_{\mathrm{D}}$ | $P_{\mathrm{LS}}$ | $P_{\mathrm{SH}}$ | SWD | GLW | OLR | HFX | LH |
|---|---|---|---|---|---|---|---|---|---|
|  |  | mm/day | mm/day | mm/day | W/m$^2$ | W/m$^2$ | W/m$^2$ | W/m$^2$ | W/m$^2$ |
| S1 | 0.56 | 12.2 | 15.7 | 4.70e-3 | 19.2 | 423 | 42.8 | 41.1 | 152 |
| S2 | 1.0 | 0 | 249 | 0 | 11.6 | 474 | 72.3 | 0.02 | 1.84 |

**Table 4.** Average values for various quantities for the S1 and S2 states from the $f_{\mathrm{LS}} \rightarrow 1$ experiment shown in Figure 11. After $f_{\mathrm{LS}}$, the three types of precipitation (deep convective $P_{\mathrm{D}}$, large-scale $P_{\mathrm{LS}}$, and shallow convective $P_{\mathrm{SH}}$) are shown, followed by the shortwave radiation to the surface (SWD), the outgoing longwave radiation from the ground (GLW), the outgoing longwave radiation at the top of the column (OLR), and the surface sensible (HFX) and latent (LH) heat fluxes.

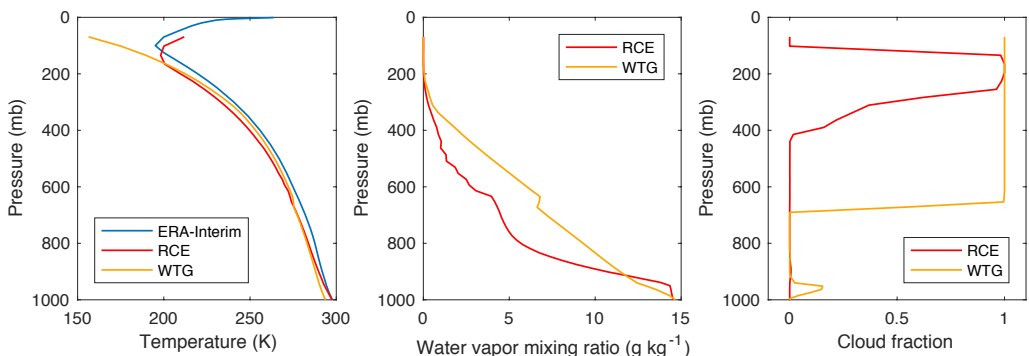

**Figure 1.** Left: Tropical soundings from ERA-Interim reanalysis (observational) data and averages over the last 30 days of two 180-day WRF SCM experiments, one a RCE experiment and one with the WTG approximation active both with fixed 300 K SST. Center: Water vapor mixing ratio profiles from the same RCE and WTG experiments. Right: Cloud fraction profiles from the same RCE and WTG experiments.



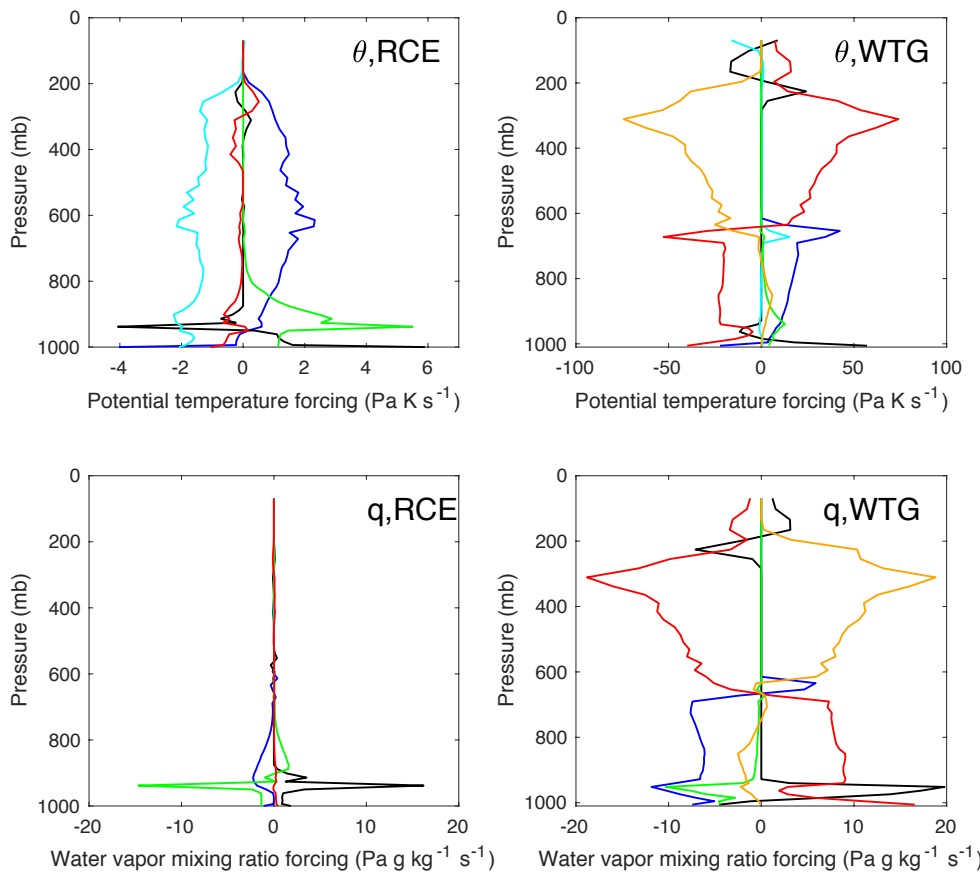

**Figure 2.** Top row: column profiles of the potential temperature ($\theta$) forcings from Eq. 1 averaged over the last 30 days of RCE (left) and WTG (right) experiments, both run for 180 days with fixed SST of 300 K. The forcings are from the deep convective (blue), shallow convective (green), boundary layer (black), radiative (cyan), and microphysics (red) CAM parameterizations and the WTG relaxation scheme (orange). Bottom row: column profiles of the water vapor mixing ratio ($q$) forcings for the same two experiments. The WTG background profile was calibrated to a SST of 300 K.



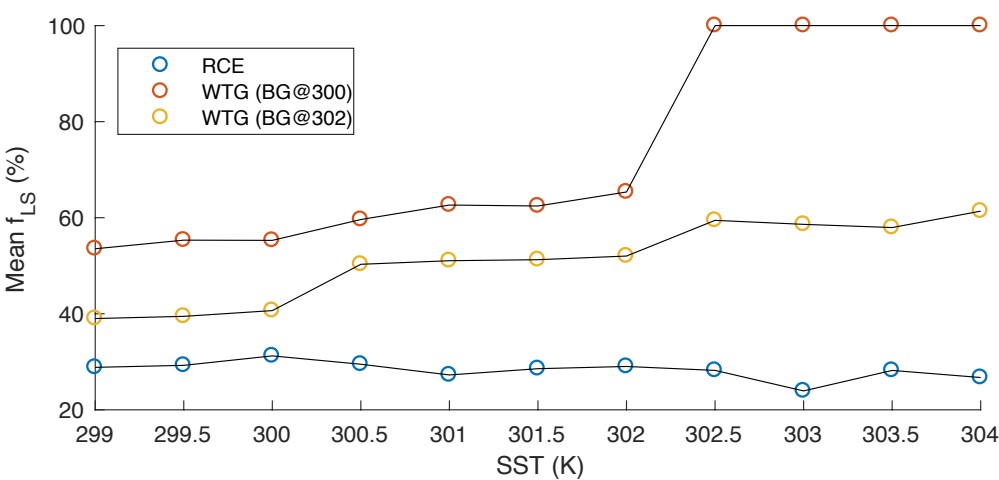

**Figure 3.** Average $f_{LS}$ for the last 15 days of a series of 30-day RCE experiments and two series of WTG approximation experiments with different background profiles, one equilibrated to a SST of 300 K and another to a SST of 302 K. In the case of the WTG experiments using the 300 K background profile, the last four experiments demonstrate the fact that $f_{LS}$ can go to unity in some cases.

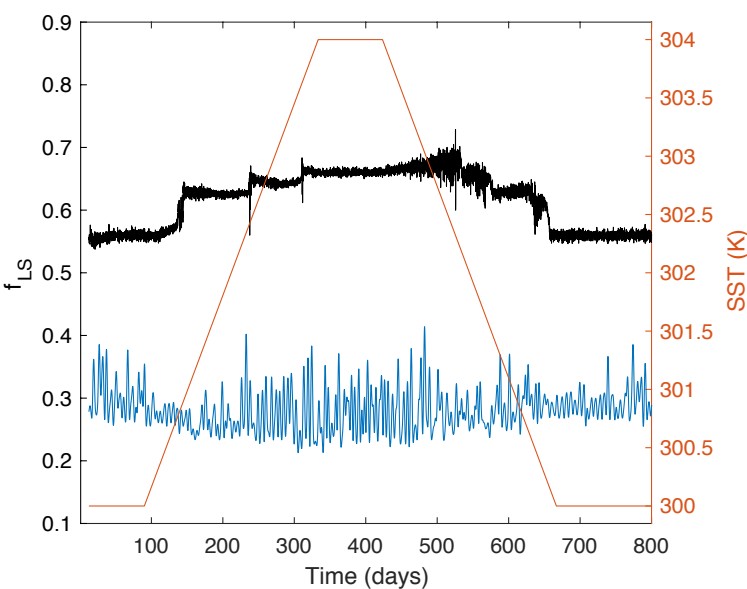

**Figure 4.** Left axis: $f_{LS}$ for the WRF SCM over 800-day RCE (blue) and WTG (black) experiments. When running the SST-forcing experiment in a RCE configuration, the unfiltered $f_{LS}$ has an average value of ∼0.28 and a standard deviation of ∼0.18. (The presented RCE $f_{LS}$ data have been low-pass filtered for clarity.) Right axis: SST over the experiments.

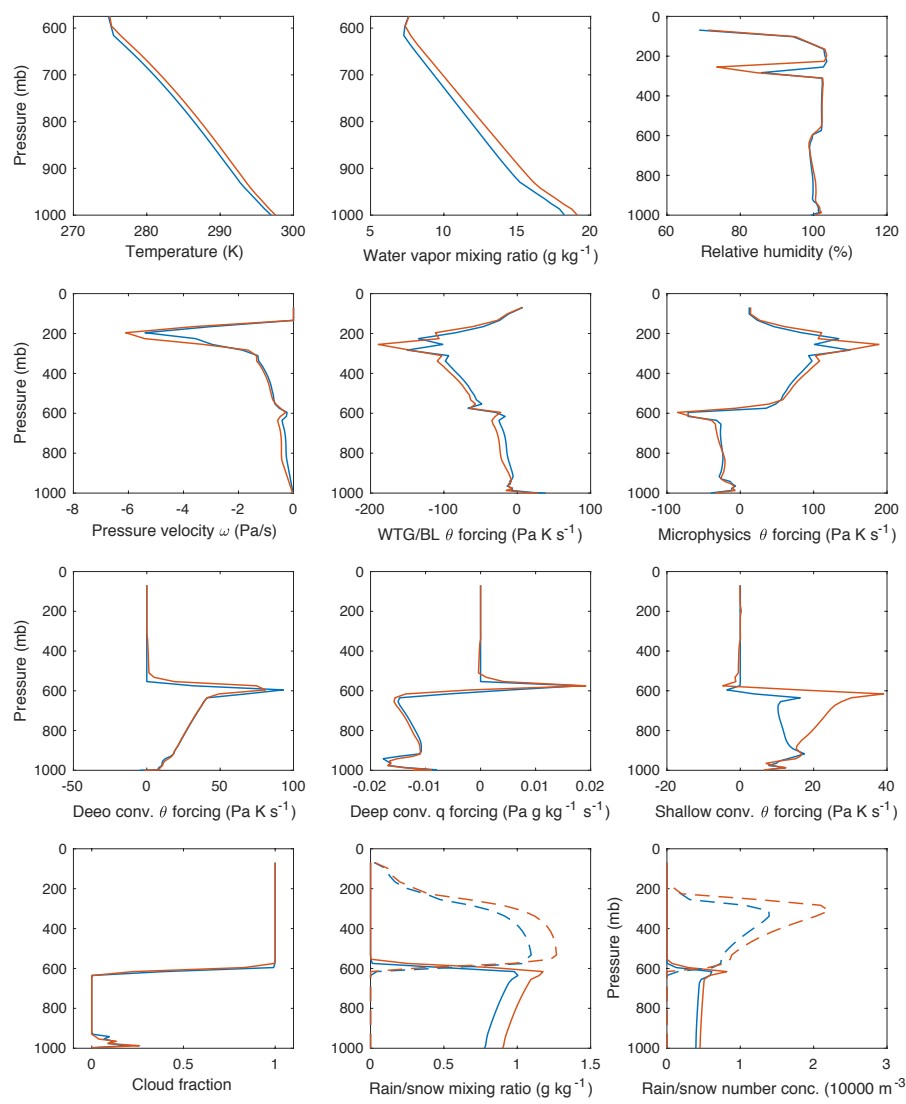

**Figure 5.** Average profiles for a number of variables for two SCM solutions at SST 304.5 K. Blue represents E1 (the equilibrium obtained during the warming phase), and red represents E2 (the cooling phase equilibrium). Forcing terms are mass-weighted in WRF, hence the units of Pa K s$^{-1}$. Note that the WTG forcing is combined with the boundary layer forcing here, but this quantity is dominated by the WTG forcing above the boundary layer. In the plots for rain/snow mixing ratio and number concentration, solid lines represent rain and dashed lines represent snow. See also Table 2.

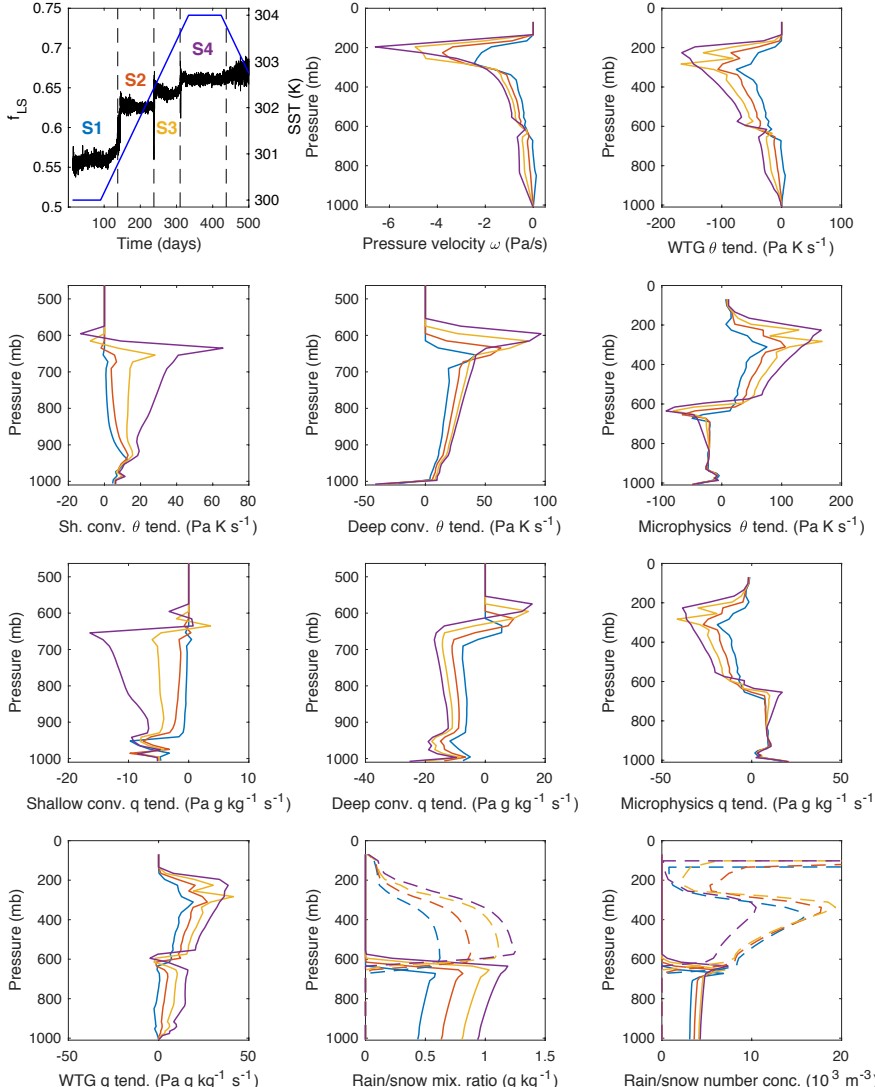

**Figure 6.** Four quasi-stationary states of the heating WRF SCM with WTG approximation. For the top left plot, the left axis shows $f_{LS}$ for the WRF SCM over the first 500 days of an 800-day integration with the WTG approximation active and a background profile calibrated for 300 K SST. Four quasi-stationary states (S1, S2, etc.) are indicated. The right axis shows SST over the same integration. The remaining plots show vertical profiles for the labeled quantities, averaged over time for each of the four states (S1 blue, S2 red, S3 yellow, S4 purple). The four plots showing convective forcing profiles stop near 500 mb because they are zero above. WRF model forcings are mass-weighted, hence the units in terms of pressure. In the last two plots for rain/snow mixing ratio and number concentration, the solid lines represent rain and the dashed lines represent snow. See also Table 3.



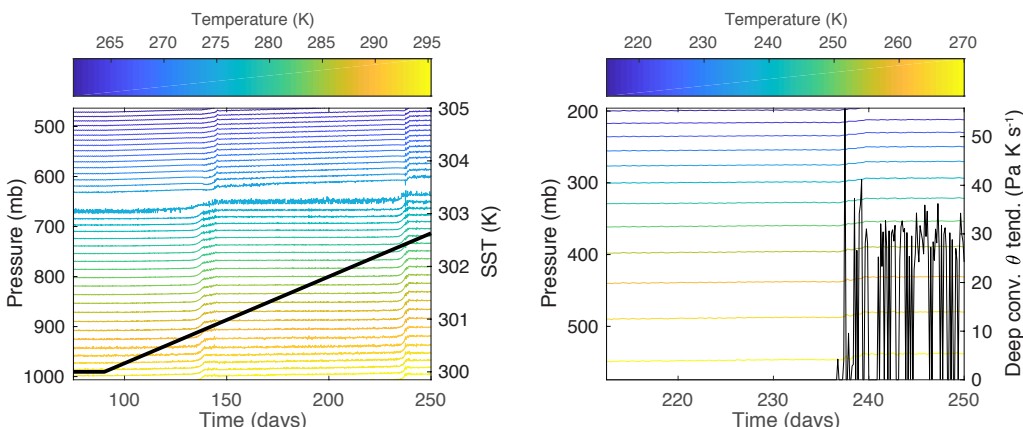

**Figure 7.** Left: A contour plot of temperature in the column, showing the first and second abrupt transitions. For reference, the SST is shown on top of the contour plot, referring to the right axis. Right: A contour plot of the upper-column temperature focusing on the second transition. On top of this contour plot, the (mass-coupled) $\theta$-forcing due to deep convection at roughly 600 mb is shown, referring to the right axis. See also Figure 9.



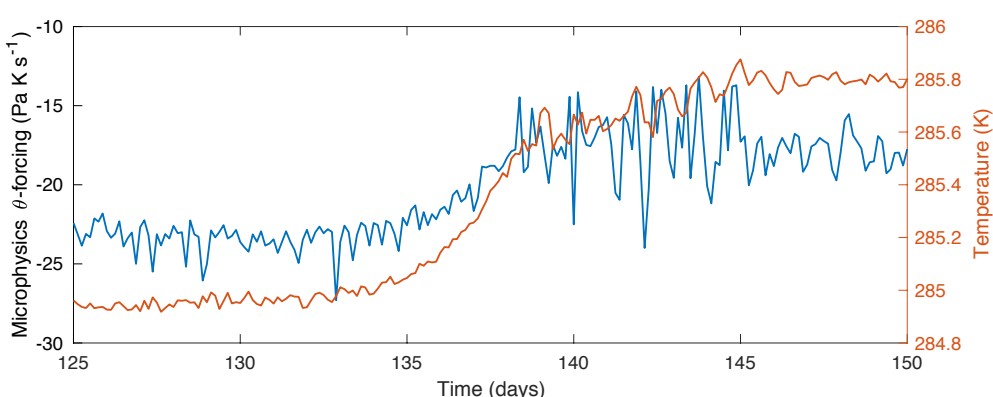

**Figure 8.** Left axis: the microphysics $\theta$-forcing at roughly 835 mb (blue) during the first abrupt transition. Right axis: the temperature at the same level (red).



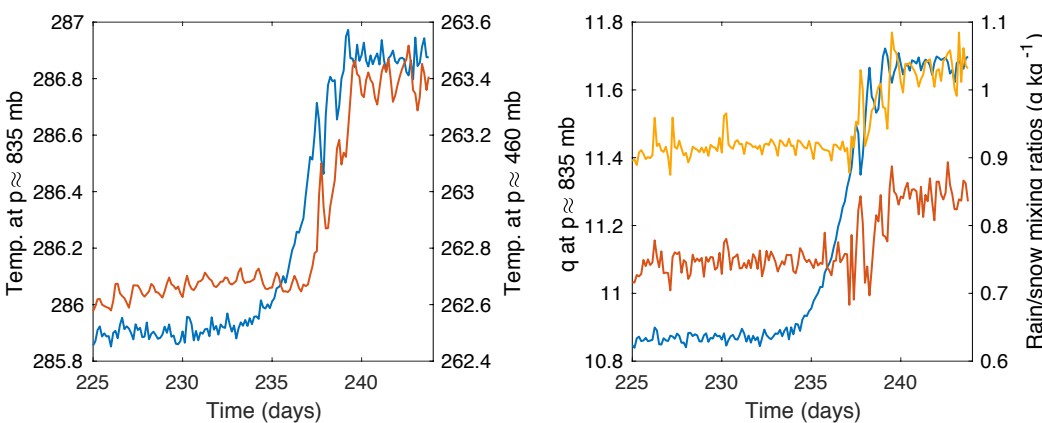

**Figure 9.** (Left panel) Left axis (blue): temperature ($T$) evolution at a randomly selected level from the lower part of the column ($p \approx 835$ mb). Right axis (red): temperature evolution at a randomly selected level from the upper part of the column (red, $p \approx 460$ mb). (Right panel) Left axis: water vapor mixing ratio ($q$) evolution at $p \approx 835$ mb (blue). Right axis: the rain (red) and snow (yellow) mixing ratios' evolution at $p \approx 835$ mb and $p \approx 460$ mb, respectively. The evolution of the latter (microphysics) variables is largely determined by the upper column.

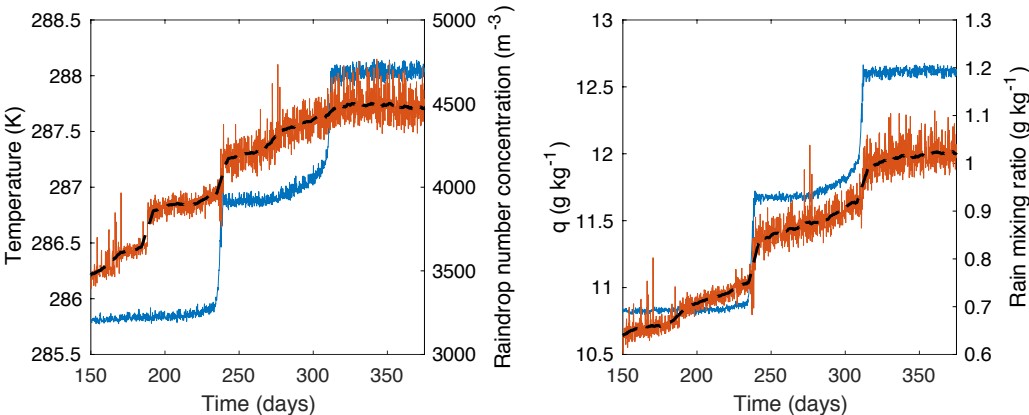

**Figure 10.** Potential precursor activity for the second and third abrupt transitions. (Left panel) Left axis: temperature ($T$) evolution (blue). Right axis: raindrop number concentration evolution (red) with low-pass filtering shown in black. (Right panel) Left axis: water vapor mixing ratio ($q$) evolution (blue). Right axis: the rain water mixing ratio evolution (red) with low-pass filtering shown in black. All quantities are at $p \approx 835$ mb.



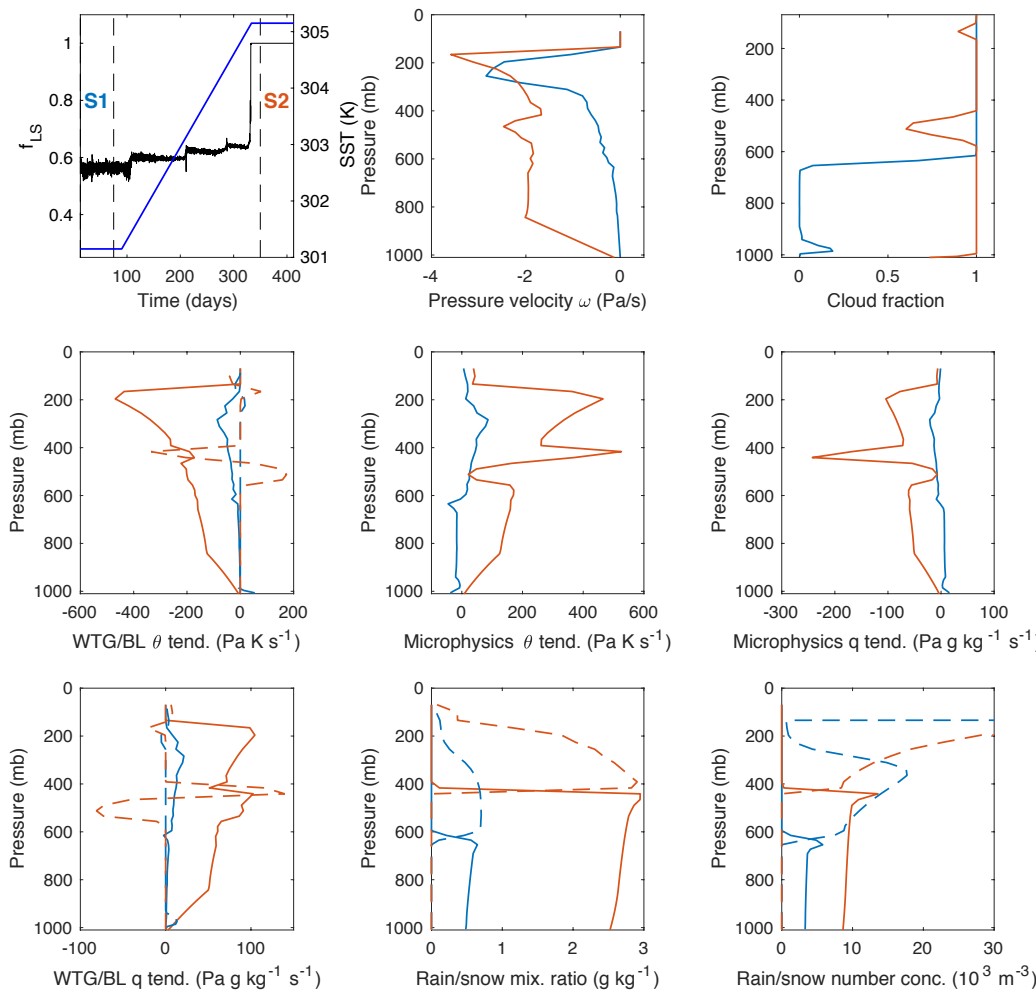

**Figure 11.** An experiment showing $f_{LS} \to 1$. The left axis of the top left plot shows $f_{LS}$ for the WRF SCM over the heating phase of an 800-day integration with the WTG approximation active and a background profile calibrated for 301.15 K SST, with SST forcing from 301.15 K (28°C) to 305.15 K. A typical quasi-stationary state (S1, blue) and the $f_{LS} = 1$ state (S2, red) are indicated. The right axis shows SST for the integration. The remaining plots show vertical profiles for the labeled quantities, averaged over time for each of the two states. In the WTG/BL panels, solid lines show the WTG relaxation forcings and dashed lines show the boundary layer forcings. In the plots for rain/snow mixing ratio and number concentration, solid lines represent rain and the dashed lines represent snow. See also Table 4.



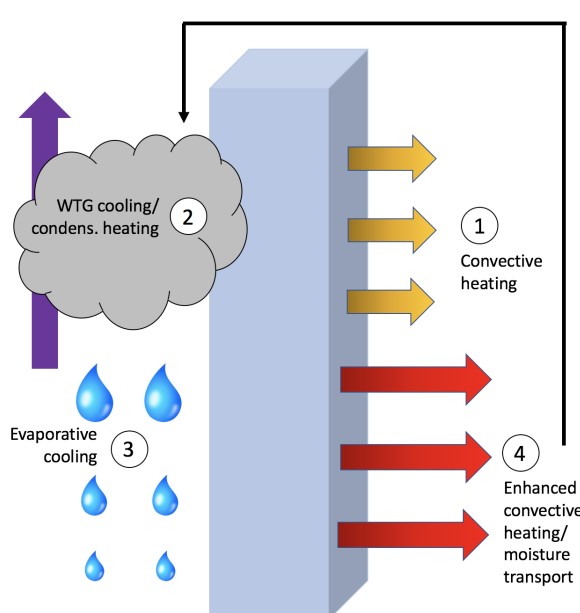

**Figure 12.** An illustration of the $\theta$-forcing balances within the WRF SCM with WTG approximation, and the order in which they appear to be established in the first few time steps. The $4 \rightarrow 2$ arrow signifies the convective parameterizations delivering moisture across the cloud base into the upper troposphere.