# Peer review of "Abrupt transitions in an atmospheric single-column model with weak temperature gradient approximation"

_Weather and Climate Dynamics, 2020_

## Referee Comment (RC1) · Anonymous Referee #1 · 18 Apr 2020

The authors study abrupt transitions with gradual increase in sea surface temperature in an idealized tropical atmosphere. The approach the authors take is the single-column modeling that incorporates the weak temperature gradient (WTG) approximation. Their numerical experiment with different SST show that abrupt transition can arise from the interplay between local deep convection and the large-scale adjustments parameterized via WTG. The authors further suggested the abrupt transition is related to evaporative cooling.

The study concerns dynamics of tropics; and it fall in the scopes of WCD. The writing is mostly clear, but needs some improvement in clarity. The language is sometimes too

informal, which needs to improve. My main concern is that the RCE experiment seem to have a closed water budget, and the WTG experiment without any SST anomalies deviate too far away from the RCE run. My comments are listed below. I recommend major revision.

Major comments:

1. The RCE experiment (Table 1) has total precipitation 1.72 mm/day (1.26+0.46), and evaporation is ∼3.42 mm/day (99.3W/m2). The two should equal to each other in RCE. But they are quite off in this RCE experiment (Table 1). Besides the water balance, there is also heat balance over the column in RCE, i.e., HFX+LH=Radiation cooling. The results indicate that the sampling period is not in RCE, and the numerical error is too large to be explained by sampling uncertainty. I suggest checking the results carefully to see if the experiments truly reach a statistical equilibrium.

2. The WTG experiment without any SST anomalies (Table 1, Fig. 1) has rain more than 200 mm/day, which is 10 times of rain in RCE. This is too large, and it's unlikely this is physical. Without any external perturbations, the WTG experiments should have rain about the same amount of rain as RCE in principle (e.g., Daleu et al. 2014). It might be that the target temperature profile is not representative, causing excessive rain in this experiment.

3. The design of the numerical experiments are sometimes quite confusing. For example, E1 and E2 (Section 3.2) are solution obtained during the warming and cooling phases. But it's never clear when the warming or cooling phases start or end, and what are the SST values in those phases. Section 3.3 discuss hysteresis and multiple equilibria, which should depend on initial conditions (dry or wet initial conditions in previous studies). But it's not clear what are the initial conditions for this experiments.

4. Section 3.3. The authors discuss abrupt transitions and quasi-stationary states. These seem to be very much dependent on the numerical configurations (e.g., whether to predict/diagnose fractional cloudy amounts). I suggest a few more ensemble runs to

ensure these abrupt transitions are robust.

Specific comments:

L4: delete "of scale"

L10-11: "re-stabilizing the lower-column evaporative cooling" does not make sense

L45-46: SCMs do have large-scale dynamics. They are in the form of prescribed large-scale forcing. What the SCMs lacks of is the interaction between the local processes in SCMs and large-scale dynamics.

L84: provide references for these CAM schemes.

L85-86: Give the reference or the URL link to the webpage.

L87: be specific how you set the model to yield fractional cloud amounts

L89-90: this discussion of cloudy points or not does not appear to be relevant to the SCM simulations, because the column is partially cloudy all the time.

L95: set latitude to 0 does not eliminate seasonal cycles in radiation. Do you still have seasonal cycles in radiation? If yes, the runs cannot really reach statistical equilibrium.

L100: this is single column experiment – only 1 grid point. Why do you need specify horizontal grid spacing?

Figures 3, 4 : f_ls is not usual state variable representing statistical equilibrium. Please also show rain or other state variables.
* * *

---

## Referee Comment (RC2) · Anonymous Referee #2 · 22 Apr 2020

This study examined the behavior of single-column simulations with weak temperature gradient approximation. As SST increases, the column shows an interesting step-wise transition toward a state in which the large-scale condensation dominates. Detailed analyses are shown to reveal the relevant processes. There are some concerns (see below).

major concerns:

1. The main feature of the state transition is the increasing role of large-scale condensation ($f\_LS$). However, this quantity is arguably an ad hoc variable due to the artificial separation of convective parameterization. If a cloud-resolving model is used instead

of a single column model, then there is not a clear separation between convective condensation and large-scale condensation. This issue makes it difficult to assess the implication and generalization of the results of this study.

2. Fig.1a The T profile of the WTG case seems strange. Where is the tropopause and stratosphere?

3. Fig. 2 It seems there is a large-scale descent below 650 hPa. Is that a common feature? It seems depends on the SCM used (e.g., Fig. 5b in Daleu). The lower level descent seems important for the later state transition.

4. section 3.3. Efforts are made to examine the differences between different states. However, an intriguing feature is the step-wise transition. May the authors examine why the transition is not gradual, but step-wise?

minor points:

Line 74. what is theta_s?

Line 84. which version of CAM?

Line 100. For a single column version of WRF, does the resolution matter? And why?

Section 4 is missing or mismarked?

---

## Author Response (AR1)

**Referee #1**

The authors study abrupt transitions with gradual increase in sea surface temperature in an idealized tropical atmosphere. The approach the authors take is the single- column modeling that incorporates the weak temperature gradient (WTG) approximation. Their numerical experiment with different SST show that abrupt transition can arise from the interplay between local deep convection and the large-scale adjustments parameterized via WTG. The authors further suggested the abrupt transition is related to evaporative cooling.

The study concerns dynamics of tropics; and it fall in the scopes of WCD. The writing is mostly clear, but needs some improvement in clarity. The language is sometimes too informal, which needs to improve. My main concern is that the RCE experiment seem to have a closed water budget, and the WTG experiment without any SST anomalies deviate too far away from the RCE run. My comments are listed below. I recommend major revision.

Regarding language informality, without specific instances it is difficult to know what changes the referee might wish to see, but we have tried to use more formal language where we could.

**Major comments:**

1. The RCE experiment (Table 1) has total precipitation 1.72 mm/day (1.26+0.46), and evaporation is  $\sim$ 3.42 mm/day (99.3W/m2). The two should equal to each other in RCE. But they are quite off in this RCE experiment (Table 1). Besides the water balance, there is also heat balance over the column in RCE, i.e., HFX+LH=Radiation cooling. The results indicate that the sampling period is not in RCE, and the numerical error is too large to be explained by sampling uncertainty. I suggest checking the results carefully to see if the experiments truly reach a statistical equilibrium.

The rainfall and evaporation actually are in balance in the RCE experiment—we had made a mistake in our calculation of the rain rates, causing them to be small by a factor of 2 (this was because our model output was printed every 3 hours but we had calculated some of the rates as if the output was every 6 hours, causing an error when converting to daily rates). We have corrected the numbers in the four Tables. This error does not affect the values of fLs, or the other variables in the Tables. Using the new numbers, evaporation equals rainfall in the RCE experiment, and as Table 1 shows, incoming shortwave radiation equals outgoing longwave radiation, consistent with statistical equilibrium. Regarding the surface fluxes, an imbalance might exist with any experiments that use fixed SSTs, but we believe the column is nevertheless in or very close to statistical equilibrium.

2. The WTG experiment without any SST anomalies (Table 1, Fig. 1) has rain more than 200 mm/day, which is 10 times of rain in RCE. This is too large, and it's unlikely this is physical. Without any external perturbations, the WTG experiments should have rain about the same amount of rain as RCE in principle (e.g., Daleu et al. 2014). It might be that the target temperature profile is not representative, causing excessive rain in this experiment.

After adjusting for the error in rainfall rates, the WTG experiment in Table 1 has a rainfall rate of ~45 mm/day (the original figure was about 20 mm/day, not 200 which may be a typo), which is at the high end of average daily tropical rainfall rates and could be more consistent with rainfall from a tropical storm for example, but is not an unphysical rate. Regarding why the control WTG experiment has more rain than the RCE experiment, we understand this as an effect of the simulated interaction with large-scale dynamics, which in this case mimics moisture convergence. There may also be a connection between the larger amounts of rainfall and the cloud-radiation interaction in the upper column—these experiments

use the realistic CAM radiation parameterization rather than prescribed radiative cooling which is common in WTG experiments (e.g. Wang and Sobel 2011, Daleu et al. 2015).

3. The design of the numerical experiments are sometimes quite confusing. For example, E1 and E2 (Section 3.2) are solution obtained during the warming and cooling phases. But it's never clear when the warming or cooling phases start or end, and what are the SST values in those phases. Section 3.3 discuss hysteresis and multiple equilibria, which should depend on initial conditions (dry or wet initial conditions in previous studies). But it's not clear what are the initial conditions for this experiments.

By "hysteresis" we mean that the history of the column matters for a solution, not just initial or boundary conditions. For the experiments described in Section 3.2, we carry out almost the same procedure as described in Section 2 with the same initial conditions (the only difference is that the warming is from 301 to 305 K instead of 300-304K, but in the revised paper we have kept the temperature increase from 300 to 304K as in the primary experiment; see the following paragraph). However, as the column is warming from 301 to 305 K, we "pause" the SST-increase at 304.5 K as described in Section 3.2, and allow the surface to remain at that SST for 30 days, then continue the warming. Then, during the cooling phase from 305 to 301 K, we again pause the temperature at 304.5 K. We find that the solutions are different for these identical boundary conditions as we go on to describe.

In the revised paper, we have made this procedure more clear and we have also used the same initial and maximum temperatures of 300 and 304 K (the same as the setup described in Section 2) for simplicity's sake. We have replaced L173+ with "To document the implied multiple equilibria, we carried out a modified version of the experiment described in Section 2, wherein we began the temperature increase at 300 K as before, paused the SST increase at a "resting" SS of 303.5 K, allowed the model to run for 30 days at the resting SST, then continued the temperature increase to 304K. We then allowed the model to rest for another 30 days at the same resting SST during the cooling phase."

4. Section 3.3. The authors discuss abrupt transitions and quasi-stationary states. These seem to be very much dependent on the numerical configurations (e.g., whether to predict/diagnose fractional cloudy amounts). I suggest a few more ensemble runs to ensure these abrupt transitions are robust.

Upon further investigation, the results discussed in this paper do not depend upon the modifications we originally made to the WRF code to ensure fractional cloud amounts—these modifications were related to an earlier experiment that made use of another set of parameterizations. Eliminating these changes to the code makes some slight differences to the rainfall rates, etc., but does not affect the qualitative behavior of the experiments. We have run new experiments with the modification to the code removed, and we have updated the numbers in our tables to reflect the new experiments as well as updated the Zenodo dataset and Zenodo reference. We have also updated Figures 1-10, but they look qualitatively similar to the previous figures. Using this new dataset eliminates the need to discuss the previous modification to the code regarding fractional cloud amounts. Moreover, as mentioned in the paper at L379+, these results are robust with the CAM physics.

Specific comments: L4: delete "of scale"

We have deleted these words.

L10-11: "re-stabilizing the lower-column evaporative cooling" does not make sense

We have reworded this to say "which allows for sufficient evaporative cooling to re-stabilize the column."

L45-46: SCMs do have large-scale dynamics. They are in the form of prescribed large-scale forcing. What the SCMs lacks of is the interaction between the local processes in SCMs and large-scale dynamics.

We have reworded this to say "SCMs generally do not account for interactions between local processes and large-scale dynamics, hence..."

L84: provide references for these CAM schemes.

We have included references for the CAM schemes.

L85-86: Give the reference or the URL link to the webpage.

We have included a reference for the technical description of WRFv3.

L87: be specific how you set the model to yield fractional cloud amounts

Consistent with our response to Comment 4 above, we have eliminated this content from the paper.

L89-90: this discussion of cloudy points or not does not appear to be relevant to the SCM simulations, because the column is partially cloudy all the time.

Consistent with our response to Comment 4 above, we have eliminated this content from the paper.

L95: set latitude to 0 does not eliminate seasonal cycles in radiation. Do you still have seasonal cycles in radiation? If yes, the runs cannot really reach statistical equilibrium.

The column is close to equilibrium as shown by the equality between incoming shortwave and outgoing longwave, as well as evaporation and precipitation (see our response to Comment 1 above). Because we are interested in the response of column physics within a realistic setting (something that might be observed within a coupled GCM, for example), our submitted draft describes experiments with realistic parameterizations, including radiation, and hence they do have a seasonal cycle. However, to be thorough and remove any change in the boundary conditions over time we have performed the same experiments with the seasonal cycle removed and we obtain qualitatively identical results. Our updated Tables and Figures are taken from these new experiments, but they look essentially the same. We also comment briefly on the similarity of the results when using prescribed radiative cooling (L382).

We have noted the lack of seasonal cycle in the new experiments (L90).

L100: this is single column experiment – only 1 grid point. Why do you need specify horizontal grid spacing?

The resolution matters in the single-column model (SCM) version of WRF, because the SCM runs on a 3x3 stencil, essentially with 9 columns (except for horizontally staggered variables like U and V), and each column identical. Since the values of dx and dy enter the WRF code even in SCM mode (for example they are used by the Zhang-McFarlane cumulus scheme), changing the horizontal resolution does affect the results, although with little effect on the transition behavior as noted at L381.

To make this more clear, we have expanded the parenthetical remark at this line to read "WRF's singlecolumn mode runs on a 3x3 stencil, hence a horizontal resolution must be specified; we chose a resolution similar to that of a typical GCM".

Figures 3,  $4 : f_{ls}$  is not usual state variable representing statistical equilibrium. Please also show rain or other state variables.

Because we believe the previous Figure 3 does not contribute significantly to the paper, and its most important features can be demonstrated with Figure 4, we have removed the previous Figure 3 from the paper. In the new Figure 3, previously Figure 4, we have included rainfall rates as requested.

**Referee #2**

This study examined the behavior of single-column simulations with weak temperature gradient approximation. As SST increases, the column shows an interesting step-wise transition toward a state in which the large-scale condensation dominates. Detailed analyses are shown to reveal the relevant processes. There are some concerns (see below).

**Major concerns:**

1. The main feature of the state transition is the increasing role of large-scale condensation (f\_LS). However, this quantity is arguably an ad hoc variable due to the artificial separation of convective parameterization. If a cloud-resolving model is used instead of a single column model, then there is not a clear separation between convective condensation and large-scale condensation. This issue makes it difficult to assess the implication and generalization of the results of this study.

While the multiple equilibria we have discovered are analyzed here in terms of the relative amounts of large-scale and convective rainfall—whose partitioning into separate parameterizations is an artifact of low resolution modeling and will not always correspond directly to the "convective" and "stratiform" rainfall of observations (Stephens et al. 2019)—we believe the equilibria observed, and the physical mechanism controlling the transitions between them, may be understood in terms of large-scale moisture convergence. The solutions observed in this SCM in fact have several similarities with features of MCSs reviewed by Houze 2018 ("100 Years of Research on Mesoscale Convective Systems"), including the higher-level heating and large amounts of rainfall. In Stephens et al. 2019, there is a discussion of the possibility that mesoscale convective systems (MCSs), which can produce large amounts of rainfall and are not explicitly represented in GCMs, might be considered an intermediate category of rainfall whose parameterization might alleviate some of the problematic aspects of using a quantity like fLs to track rainfall behavior.

2. Fig.1a The T profile of the WTG case seems strange. Where is the tropopause and stratosphere?

While we cannot give a complete account of the shape of the T profile in Fig 1, we believe the profile may be related to our use of a realistic radiation parameterization in combination with the WTG approximation, considering that an important difference between these and other WTG experiments—such as those described in Daleu et al. 2015—is that WTG studies often employ some form of prescribed radiative cooling, rather than a standard radiation parameterization, to keep temperatures near desired values in this region. In Daleu et al. 2015, for example, idealized radiative cooling was used to maintain "the temperature of the upper troposphere and stratosphere at a uniform value of 200 K").

3. Fig. 2 It seems there is a large-scale descent below 650 hPa. Is that a common feature? It seems depends on the SCM used (e.g., Fig. 5b in Daleu). The lower level descent seems important for the later state transition.

WTG models show a variety of vertical wind behavior as demonstrated by Daleu et al. 2015, as the referee states. In this case, we believe the interaction between the column and the simulated large-scale dynamics moisture convergence, and the WTG scheme works to offset evaporative cooling in the lower column by inducing a small large-scale descent. Once the deep and shallow convective schemes start to heat the lower column more, as the SST increase begins, WTG switches into a cooling role in the lower column. The WTG behavior does adjust at each transition, with the lower-column pressure velocity decreasing (i.e. becoming more negative) at each transition, but we believe this can only be a response to, not a cause of, the stepwise transitions, which we believe are ultimately caused by nonlinearities within the Morrison-Gettelman microphysics.

4. section 3.3. Efforts are made to examine the differences between different states. However, an intriguing feature is the step-wise transition. May the authors examine why the transition is not gradual, but step-wise?

Our discussion in Section 3.4 of the "runaway" solution, where all rainfall is large-scale, hints at the possibility that the WTG solutions in these experiments are only kept from showing the  $f_{LS} = 1$  behavior by the action of evaporation in the lower column. As this evaporation starts to fail, the column adjusts to a new quasi-stationary behavior. We believe this dependence on the lower-column evaporation limits the effective degrees of freedom of the system so that its global behavior is dictated by the details controlling evaporation rates, namely nonlinearities within the Morrison-Gettelman microphysics. These nonlinearities are what cause the sudden transitions within the column rather than more gradual behavior.

Minor points:

Line 74. what is theta\_s?

"Theta\_s" was an artifact of the dissertation from which this paper originated and should have been edited. We have replaced the symbol "theta\_s" with the words "the quasi-stationary state temperature".

Line 84. which version of CAM?

We have noted this version of WRF is based on CAM 3.0.

Line 100. For a single column version of WRF, does the resolution matter? And why? Section 4 is missing or mismarked?

The resolution matters in the single-column model (SCM) version of WRF, because the SCM runs on a 3x3 stencil, essentially with 9 columns (except for horizontally staggered variables like U and V), and each column identical. Since the values of dx and dy enter the WRF code even in SCM mode (for example they are used by the Zhang-McFarlane cumulus scheme), changing the horizontal resolution does affect the results, although with little effect on the transition behavior as noted at L381.

To make this more clear, we have expanded the parenthetical remark at this line to read "WRF's singlecolumn mode runs on a 3x3 stencil, hence a horizontal resolution must be specified; we chose a resolution similar to that of a typical GCM". Section 4 is not missing or mismarked, the label is at the top of page 10.

**Additional significant changes:**

1. Along with removing the previous Figure 3, we have removed the previous Figure 7 from the paper, since we believe the points it illustrates can be made with reference to other figures and/or described adequately in the text alone.

[revised manuscript text omitted]
 | 0.28 0.29             | <del>1.26</del> -2.57 | 0.46-0.83            | 2.614.63e-4       | <del>270-276</del>    | <del>358_354</del>  | <del>272-277</del>    | <del>8.94 9.01</del> | <del>99.3-97.7</del> |
| WTG | <del>0.56</del> -0.62 | <del>9.55-</del> 27.3 | <del>12.1-45.4</del> | 2.08.6e-3         | <del>14.6</del> -13.4 | <del>416-</del> 409 | <del>34.9-</del> 36.4 | <del>48.2</del> 53.0 | <del>165</del> -186         |

**Table 1.** A series of quantities averaged over the last 30 days from 180-day RCE and WTG approximation experiments without anomalous SST forcing. After  $f_{LS}$ , the three types of precipitation (deep convective  $P_D$ , large-scale  $P_{LS}$ , and shallow convective  $P_{SH}$ ) are shown, followed by the shortwave radiation to the surface (SWD), the outgoing longwave radiation from the ground (GLW), the outgoing longwave radiation at the top of the column (OLR), and the surface sensible (HFX) and latent (LH) heat fluxes.

|    | $f_{\rm LS}$          | $P_{\rm D}$ $P_{\rm LS}$ |                       | $P_{\rm SH}$              | P SH SWD           |                               | HFX                  | LH                  |
|----|-----------------------|--------------------------|-----------------------|---------------------------|-------------------------------|-------------------------------|----------------------|---------------------|
|    | 0                     | mm/day                   | mm/day                | mm/day                    | $W/m^2$                       | $W/m^2$                       | $W/m^2$              | $W/m^2$             |
| E1 | <del>0.61</del> -0.64 | <del>42.4 41.4</del>     | <del>66.0</del> -74.3 | <del>0.026-</del> 3.35e-2 | <del>12.9</del> - 11.5 | <del>39.9</del> - 38.4 | 67.3                 | <del>267_252</del>  |
| E2 | <del>0.63</del> 0.66  | <del>45.4 4</del> 4.0    | <del>77.8</del> 86.8  | 0.056-5.63e-3             | <del>11.4</del> -11.8         | <del>40.7-39</del> .1         | <del>57.3</del> 58.1 | <del>231-</del> 221 |

**Table 2.** Average values for a series of quantities for the two distinct SCM equilibria (E1 and E2) shown in Figure 4. After  $f_{LS}$ , the three types of precipitation (deep convective  $P_D$ , large-scale  $P_{LS}$ , and shallow convective  $P_{SH}$ ) are shown, followed by the shortwave radiation to the surface (SWD), the outgoing longwave radiation at the top of the column (OLR), and the surface sensible (HFX) and latent (LH) heat fluxes.

|            | $f_{LS}$              | $P_{D}$                     | $P_{\rm LS}$                | $P_{\rm SH}$              | SWD                          | GLW                | OLR                   | HFX                         | LH                  |
|------------|-----------------------|-----------------------------|-----------------------------|---------------------------|------------------------------|--------------------|-----------------------|-----------------------------|---------------------|
|            |                       | mm/day                      | mm/day                      | mm/day                    | $W/m^2$                      | $W/m^2$            | $W/m^2$               | $W/m^2$                     | $W/m^2$             |
| S 1 | 0.56                  | <del>9.58-19.4</del> | <del>12.2</del> 24.9        | <del>2.1</del> 4.3e-3     | <del>17.6-</del> 15.2 | <del>416_406</del> | <del>35.6-35.0</del>  | <del>48.3 47.5</del>        | <del>165_163</del>  |
| S2         | <del>0.63-</del> 0.62 | <del>14.6 28.7</del> | <del>24.5 47.6</del>        | <del>5.3</del> 9.9e-3     | <del>13.3-13.0</del>  | <del>412_410</del> | 36.5                  | <del>58.5 57.2</del>        | <del>207_202</del>  |
| S 3 | 0.64                  | <del>19.5 38.6</del> | <del>35.4 68.9</del> | <del>14.8e-3-2.8e-2</del> | <del>14.0-12.6</del>  | 416                | <del>38.0-37.9</del>  | <del>63.7 63.3</del> | <del>235_234</del>  |
| S 4 | 0.66                  | 23.4-46.4                   | <del>45.7-89</del> .1       | <del>31.9e-3</del> 6.0e-2 | <del>13.4</del> -12.2        | 422                | <del>39.5</del> -39.4 | <del>63.2</del> 63.5        | <del>243-</del> 245 |

S4 0.66 23.4.46.4 45.7-89.1 31.9e-3.6.0e-2 13.4-12.2 422 39.5-39.4 63.2-63.5 243-245. **Table 3.** Average values for a series of quantities for the four states from the experiment shown in Figure 5. After  $f_{LS}$ , the three types of precipitation (deep convective  $P_D$ , large-scale  $P_{LS}$ , and shallow convective  $P_{SH}$ ) are shown, followed by the shortwave radiation to the surface (SWD), the outgoing longwave radiation from the ground (GLW), the outgoing longwave radiation at the top of the column (OLR), and the surface sensible (HFX) and latent (LH) heat fluxes.

|            | $f_{LS}$             | $P_{\rm D}$                 | $P_{LS}$             | $P_{\rm SH}$ | SWD                          | GLW                 | OLR                           | HFX       | LH                          |
|------------|----------------------|-----------------------------|----------------------|--------------|------------------------------|---------------------|-------------------------------|-----------|-----------------------------|
|            |                      | mm/day                      | mm/day               | mm/day       | $W/m^2$                      | $W/m^2$             | $W/m^2$                       | $W/m^2$   | $W/m^2$                     |
| S 1 | <del>0.56</del> 0.55 | <del>12.2</del> 24.3 | <del>15.7-30.3</del> | 4.708.0e-3   | <del>19.2-</del> 17.1 | <del>423-415</del>  | <del>42.8</del> - 42.6 | 41.1-40.2 | <del>152</del> - 149 |
| S2         | 1.0                  | 0                           | <del>249</del> -412  | 0            | <del>11.6 9</del> .8         | <del>474</del> -458 | <del>72.3-</del> 57.6         | 0.02 0.03 | <del>1.84</del> -1.77       |

**Table 4.** Average values for various quantities for the S1 and S2 states from the  $f_{LS} \rightarrow 1$  experiment shown in Figure 9. After  $f_{LS}$ , the three types of precipitation (deep convective  $P_D$ , large-scale  $P_{LS}$ , and shallow convective  $P_{SH}$ ) are shown, followed by the shortwave radiation to the surface (SWD), the outgoing longwave radiation from the ground (GLW), the outgoing longwave radiation at the top of the column (OLR), and the surface sensible (HFX) and latent (LH) heat fluxes.

**Figure 1.** Left: Tropical soundings from ERA-Interim reanalysis (observational) data and averages over the last 30 days of two 180-day WRF SCM experiments, one a RCE experiment and one with the WTG approximation active both with fixed 300 K SST. Center: Water vapor mixing ratio profiles from the same RCE and WTG experiments. Right: Cloud fraction profiles from the same RCE and WTG experiments.